# Age-Associated Increase in Thrombogenicity and Its Correlation with von Willebrand Factor

**DOI:** 10.3390/jcm10184190

**Published:** 2021-09-16

**Authors:** Parnian Alavi, Abhisha M. Rathod, Nadia Jahroudi

**Affiliations:** Department of Medicine, University of Alberta, Edmonton, AB T6G 2S2, Canada; palavi@ualberta.ca (P.A.); abhisha@ualberta.ca (A.M.R.)

**Keywords:** von Willebrand factor, endothelial cells, aging, thrombosis, platelet aggregates, ADAMTS13, COVID-19

## Abstract

Endothelial cells that cover the lumen of all blood vessels have the inherent capacity to express both pro and anticoagulant molecules. However, under normal physiological condition, they generally function to maintain a non-thrombogenic surface for unobstructed blood flow. In response to injury, certain stimuli, or as a result of dysfunction, endothelial cells release a highly adhesive procoagulant protein, von Willebrand factor (VWF), which plays a central role in formation of platelet aggregates and thrombus generation. Since VWF expression is highly restricted to endothelial cells, regulation of its levels is among the most important functions of endothelial cells for maintaining hemostasis. However, with aging, there is a significant increase in VWF levels, which is concomitant with a significant rise in thrombotic events. It is not yet clear why and how aging results in increased VWF levels. In this review, we have aimed to discuss the age-related increase in VWF, its potential mechanisms, and associated coagulopathies as probable consequences.

## 1. Introduction

A major contributing factor to aging-associated morbidity and mortality is the increased prevalence of thrombogenicity and consequently thrombotic disorders [1]. Thrombosis is the formation of blood clots within a blood vessel, and complications of thrombosis may cause acute life-threatening or chronic conditions [2]. Thrombotic events in large vessels, such as coronary arteries and cerebral blood vessels, could lead to acute catastrophic events, for instance, heart attack and stroke respectively. Moreover, microvascular thrombosis may contribute to the target organ(s)’ suboptimal functioning as a result of perfusion hindrance. Microvascular thrombosis is the hallmark of a number of vascular diseases, including thrombotic microangiopathy, disseminated intravascular coagulation, and sickle cell disease among others [2,3]. Although many genetic and environmental risk factors contribute to thrombotic events in the general population, aging is a universal and non-modifiable risk factor for thrombosis [4]. Epidemiological studies have indicated an association between age and higher risk incidents of thrombotic cardiovascular complications, including acute myocardial infarction, stroke, and venous thromboembolism [5,6,7]. Aging considerably increases the incidence of venous and arterial thrombotic events, with an annual rate of venous thromboembolism (VTE) from 1 per 10,000 in young, to 1 in 100 among the elderly population over the age of 80 [8]. It has been reported that after the age of 45, the incidence of atherothrombotic brain infarction and myocardial infarction doubles after every 10 years of life [7,9]. The cause of this sharp increase in thrombosis with aging is not yet clearly understood.

The dramatic increase in thrombogenicity in the elderly population may reflect the pathophysiology of aging. Aging-associated development of molecular and anatomical abnormalities have been detected that could contribute to this process. These include increase in diameter and thickness of vessel walls, fragmentation of internal elastic lamina, and vascular smooth muscle cells hypertrophy within the vessel walls that results in endothelial injury, which is a key factor in promoting thrombosis [10,11,12,13]. Aging is also associated with the development of a pro-inflammatory phenotype, a well-established risk factor for thrombosis [14,15,16]. Furthermore, several clinical studies have reported that aging alters the levels of various circulating and vessel wall-associated factors, leading to an imbalance in the ratio of procoagulants to anticoagulants in favour of procoagulants, and consequently a prothrombotic state [17,18].

Procoagulant protein von Willebrand factor (VWF) is among factors that display an age-associated increase in circulation [19]. VWF is fundamental to the process of thrombus generation since it initiates the first step of the process, specifically adherence of platelets to endothelial/subendothelial surfaces [20]. It also mediates the subsequent step of platelet-platelet interactions that is necessary for the formation of platelet aggregates [20]. The circulating levels of VWF are not only elevated with aging in healthy population, but are also influenced by multiple pathologies associated with the process of aging, such as insulin resistance, obesity, and various risk factors that are associated with cardiovascular disorders [21,22]. Thus, it stands to reason that increased production of VWF, without a proportional increase in anticoagulant proteins, may contribute significantly to increased thrombogenicity with aging [16,19]. In this review, we aim to discuss the impact of aging on the VWF levels, its consequences under physiological/pathophysiological conditions, and potential mechanism(s) that may instigate aging-associated elevation of VWF.

## 2. VWF Role in Physiology

VWF is an adhesive multimeric glycoprotein that plays a pivotal role in normal hemostasis and thrombosis [23]. It was first identified by Dr. Eric von Willebrand during the treatment of a severe bleeding disorder [24]. The gene coding for VWF is located on chromosome 12, spans 180 kb and contains 52 exons [25,26]. VWF expression is strictly restricted to two cell types, endothelial cells (ECs) and megakaryocytes (precursor of platelets) [25]. VWF biosynthesis is a complex process that starts with the translation of an approximately 9 kb VWF mRNA to generate a precursor protein that is subjected to a wide range of post-translational modifications. These include dimerization, multimerization, and cleavage, in addition to extensive glycosylation and sulfation [27,28]. Cleavage of the precursor protein generates two components, pro-peptide (VWFpp) and mature subunit (VWF), which remain non-covalently associated until secreted into the bloodstream [27]. Multimerized VWF is stored in specialized organelles known as Weibel-Palade bodies (WPB) and alpha granules in endothelial cells and platelets, respectively [27]. While alpha-granules release VWF upon activation of platelets, endothelial cells release VWF by two pathways, namely constitutive and regulatory secretions. Dimer/smaller multimers of VWF are secreted continuously from endothelial cells (consecutive secretion), while most biologically active ultra-large WPB-stored multimers (UL-VWF) are secreted through regulatory pathway in response to inducers [29]. Biosynthesis, structure, processing, and secretion of VWF have been subjects of many excellent reviews periodically [27,29,30].

VWF that is secreted from endothelial cells either enters circulation or is deposited in the subluminal extracellular matrix, which when exposed in response to injury, provides the binding sites for platelets [2,6]. Due to its highly adhesive nature that allows association with various extracellular matrix proteins, circulating VWF can also interact with other proteins (such as collagen) in exposed extracellular matrix at the site of vascular injury [31]. When the shear rate is elevated, globular VWF is unfurled and its various binding sites are exposed, mediating a transient interaction between VWF and platelet [32]. This leads to initial platelet captures and slows platelets movement, which is then stabilized through the interaction of various other proteins, in addition to VWF (Figure 1) [33,34]. Once initially captured platelets are stabilized and activated to release their VWF, VWF and other proteins on adherent activated platelets provide recruitment and attachment sites for additional circulating platelets [35,36]. Activated platelets provide a major site for prothrombin complexes assembly and enhance thrombin generation, thus directly participating in thrombosis. This physiological function of subluminal VWF is critical for maintaining vessel integrity while the repair of the vessel wall proceeds at the site of injury. Upon completion of the repair process, VWF cleaving enzymes, mainly metalloproteinase ADAMTS13 (although other proteases that target VWF, i.e., ADAMTS28, have also been identified [37]), as well as other thrombolytic enzymes, mediate the disassembly of platelet aggregates, leading to thrombus resolution and restoration of unobstructed blood flow (Figure 1). However, VWF is also shown to bind to the endothelial cell membrane in the intact luminal surface [38]. Membrane-bound VWF could also mediate platelet adhesion to the intact endothelial surface under permissive conditions (Figure 1), and consequently, this may be a contributing factor towards undesired thrombus formation [38].

VWF also performs additional hemostatic functions by binding and stabilizing FVIII (a vital coagulation protein of the intrinsic clotting pathway) in circulation. Furthermore, VWF has been recently shown to participate in a number of other physiological and pathophysiological processes, such as inflammation, angiogenesis, and cancer metastasis, which are the subject of many recent excellent reviews and will not be discussed here [39,40,41,42].

Plasma levels of VWF are influenced by a combination of environmental, genetic, and pathological factors, although what is considered as normal levels constitute a wide range (50–200%) in a healthy population [35]. It is estimated that about 30% of the variation in the plasma VWF levels is related to genetic factors through pedigree analyses, while twins’ studies have suggested this value to be 65% [35,36,43,44]. Another major contributing factor to the variation in circulating VWF levels is the ABO blood group, which is proposed to account for approximately 30% of the observed variability [35,45]. Blood Group O individuals have 25% lower plasma VWF levels than non-O blood group individuals [45,46]. The presence of AB antigens coincides with a protective function against VWF clearance. This is proposed as the main reason for higher VWF levels in the non-O blood group; although recent evidence suggesting a significantly decreased levels of cellular VWF protein in lung endothelial cells of the O blood group individuals compared to non-O have also been presented [47].

## 3. Coagulopathies Associated with von Willebrand Factor

Considering the central role of VWF in hemostasis and thrombosis, both decreased and increased levels of VWF (outside the normal range) are associated with coagulopathies. Deficiencies of VWF, either quantitative or qualitative, are associated with the most common inherited bleeding disorder known as Von Willebrand disease (VWD) [48,49,50]. There are four main types of VWD based on phenotypic analysis of the VWF: Type 1, Type 2, Type 3, and platelet-type [51,52,53]. The most common and mildest form of VWD is Type 1 (quantitative) that is caused by low VWF production or accelerated clearance [53]. Type 2 VWD (qualitative) is characterized by defects in VWF (abnormal function), and it accounts for 20–40% of VWD cases [51]. Whereas Type 3 (qualitative) VWD is a total quantitative deficiency of VWF and is the most severe form of VWD that represents 5% of VWD cases [53]. In platelet-type VWD, a gain of function mutation within platelet surface receptor for VWF leads to exaggerated platelet-VWF interaction, resulting in thrombocytopenia and consequently prolonged bleeding time [52].

In contrast to VWF deficiency, increased plasma VWF levels and functional activity is a significant risk factor for thrombosis [54]. The thrombotic event may involve large vessels, leading to acute consequences such as stroke or heart attack [55]; or implicate microvasculature, leading to various chronic diseases such as thrombotic microangiopathy [56]. Significantly elevated levels of VWF are observed in thrombotic thrombocytopenic purpura (TTP), which results from a profound deficiency of ADAMTS13. It is the leading cause of thrombotic microangiopathy characterized by the formation of platelets aggregates in arterioles, capillaries, and venules [57,58]. Several other pathological conditions including vascular dementia [58,59], Alzheimer’s disease [59,60], traumatic brain injury [61], acute respiratory distress syndrome [62], and kidney failure [63] are also associated with increased plasma levels of VWF. Such disorders may arise or be exacerbated potentially as a consequence of increased microvascular occlusion.

## 4. Influence of Aging on VWF and ADAMTS13 in a Healthy Population

Age-associated increases in plasma levels of VWF have been reported in studies using animal models [64,65,66], but also well-established in the human population (Table 1). A study of 74 centenarians and 110 individuals aged 45–86 years old has shown that VWF antigen levels, and its functional activity, were markedly elevated in centenarians compared to the younger cohort [67]. The study did not find any significant difference between Blood Group O and non-O [67]. Another cross-sectional study on a cohort of 207 individuals aged 1–87 years demonstrated that the levels of circulating VWF increased significantly in the elderly (with an overall increase of 1.56-fold), and that the increase was more significant in the non-O blood group individuals [68]. Although the influence of the ABO blood group was not significant in young individuals, it became more pronounced with advancing age [68]. Furthermore, the age-related increase in VWF levels was accompanied by a 2.03-fold increase in VWF functional activity in older individuals [68]. This increase in VWF levels was followed by simultaneous elevation in FVIII coagulant activity and antigen levels (reaching up to 1.5-fold by later life in healthy individuals) [68,69]. A recent cross-sectional study of 2923 individuals also reported elevation of VWF:Ag as well as FVIII coagulation activity per decade of age after adjustment for comorbidities including body mass index, inflammation and hormone use; and the increase was higher for non-O blood group individuals [69]. However, the influence of aging and plasma VWF levels on FVIII level and coagulant activity in mild Hemophilia A patients is scarcely studied. One study reported a significant positive correlation between FVIII coagulant activity and aging without amelioration of bleeding [70]; whereas another study identified only minor influence of aging on FVIII levels/activity [71]. A cross-sectional study of 3616 Japanese participants, aged 30 to 79 years reported that VWF levels tend to increase (significantly influenced by ABO blood groups), whereas ADAMTS13 activity was decreased (not significantly influenced by ABO blood groups) with advancing age, leading to an increased ratio of plasma VWF levels -to-ADAMTS13 activity with aging in both men and women [72]. It is worth noting that this study reported a lower level of ADAMTS13 activity in men compared to women, suggesting that elderly males are at a higher risk of developing hypercoagulable state and consequently thrombosis [72]. Another study of the healthy Arab population revealed that VWF levels increase with aging, while ADAMTS13 antigen levels remain unaffected but ADAMTS13 activity decreases with aging [73]. Inconsistent findings have been reported concerning the impact of biological sex on rising VWF levels with aging. The large study of blood donors in South Wales (n = 5052) reported that there was a minor but significant difference between VWF levels among men and women (approximately 114 IU ${\mathrm{dL}}^{-1}$ for men vs. 109 IU ${\mathrm{dL}}^{-1}$ for women) [73]. In contrast, the study of the healthy Arab population found lower plasma levels of VWF in females compared to males, but only in Blood Group O females [73]. ADAMTS13 activity and antigen levels were unaffected by blood groups in both genders of the Arab population [73]. Collectively these reports suggest that the trend of rising VWF levels with aging is similar in both women and men. While it is reported that women have a higher rate of increase in the VWF plasma levels during middle age (41–50 years), this may be attributed to changes in hormone levels during pregnancy, menstrual cycle, hormone-based contraceptives, and menopause, all of which can influence VWF levels [74,75,76].

From several studies, it is clearly demonstrated that circulating VWF levels tend to increase gradually with aging in humans, at a 1–2% rate per year in adults, but the absolute increase is greater with advancing age [72,77,78]. However, a higher rate of elevation in VWF levels with age occurs after midlife (above 40 years of age) in both men and women [21,74]. It is reported that among different variables such as age, gender, blood type, and use of medications, only age was significantly predictive of VWF level, and exhibited age-related elevation [74]. Since VWFpp and mature VWF have different half-lives (2 h and around 8–12 h for each respectively), VWFpp levels have been used as a marker for endothelial cell activation and VWF release [68,79]. VWFpp and mature VWF are cleared independently, hence multiple studies have utilized the ratio between VWFpp and VWF antigen (VWF:Ag) levels to evaluate the synthesis and clearance of VWF [80,81]. These analyses have demonstrated that in addition to upregulated VWF secretion, a reduction in the ratio of the VWFpp:VWF antigen is observed with aging, suggesting an age-associated decrease in VWF clearance and consequently its increased half-life in circulation [68]. However, a report of immunohistochemical analysis of cellular VWF expression in lung tissues of 64 pulmonary neoplasia patients [26 children (mean age 3.5 years) and 38 adults (mean age 55 years)], demonstrated a significantly elevated levels of cellular VWF in arterioles, capillaries, and venules of adult lungs compared to children [82]. This observation suggests that in addition to increased secretion and decreased clearance, a potential increase in VWF biosynthesis, specifically in smaller vessels, maybe a significant contributing factor to the elevated VWF levels in response to aging. Furthermore, another potential source of the age-associated increase in VWF may involve alteration in platelets activity and consequently increased release of VWF from platelets alpha-granules. For instance, aging is associated with increased content of basal phosphoinositide in the platelet plasma membrane [83], which is important for activating the second messenger signaling pathway and triggering platelet activity. This suggests that aging may alter platelet transmembrane signaling pathways and consequently its propensity for activation and as a result platelet aggregate formation. Furthermore, studies have shown an age-related reduction in nitric oxide (NO) levels [84,85,86], contributing to abnormal platelet aggregation. Collectively, procoagulant alterations in endothelial cells’ phenotype and platelets activities with aging, may contribute to enhanced VWF levels and decreased ADAMTS13 activities, providing the foundation for thrombosis.

**Table 1 jcm-10-04190-t001:** Effect of age on Von Willebrand factor in the healthy population.

Location	Population Characteristics	Results	Author
Italy	184 healthy individuals included 74 centenarians (100–107 years), 55 younger controls (<45 years), 55 older controls (>45 years)	centenarians VWF:Ag level 245 (U/dL) in O blood group and 285 (U/dL) in non-O blood group; older controls VWF:Ag level 99 (U/dL) in O blood group and 152 (U/dL) in non-O blood group; younger controls VWF:Ag level 77 (U/dL) in O blood group and 115 (U/dL) in non-O blood group; VWF activity level centenarians > older controls > younger controls	Coppola et al., 2003 [67]
Canada	207 healthy individuals included 113 old (55–87 years), 42 middle-age (30–49 years), 52 young (1–17 years)	Plasma VWF level increased with age reaching a 1.71-fold by old age in non-O and 1.25-fold in O blood group. VWF activity reached 2.03-fold by old age and VWFpp level (as a marker of VWF secretion) elevated to 1.26-fold in older individuals with blood type non-O than blood type O.	Albanez et al., 2016 [68]
United Kingdom	Cohort of 5052 healthy individuals	VWF:Ag level increase minor and non-significant up to age of 40 years but elevated significantly above age >40 years. For the whole cohort absolute increase between each age group: 2 IU/dL between <20 years and 21–30 years, 3 IU/dL between 21–30 years and 31–40 years, 7 IU/dL between 31–40 years and 41–50 years, 8 IU/dL between 41–50 years and 51–60 years, and 15 IU/dL between 51–60 years and 61–70 years.	Davies et al., 2012 [74]
Japan	Cohort of 3616 healthy Japanese population between the age range of 30–79 years	VWF:Ag increased with advancing age and the linear regression coefficient being 1.37 and 1.30 in men and women respectively. Plasma ADMATS13 activity decreased with age, significantly after 60 years and regression coefficient was −0.642 resulted in increased ratio of VWF:Ag-to-ADAMTS13 activity with age.	Kokame et al., 2011 [72]
Kuwait	200 healthy individuals	Plasma VWF level significantly higher in older individuals and ADAMTS-13 activity decreased with age, however, ADAMTS13 level not affected by age.	Al-Awadhi et al., 2014 [73]
The Netherlands	Cohort of 2923 individuals between the age range of 18–70 years	VWF:Ag increase per decade of age 18 IU/dL and for FVIII activity 12 IU/dL. After adjustment for acquired factors (comorbidities, body mass index, reduced kidney function, hormone use, and inflammation), the increase per decade 13 IU/dL for VWF:Ag and 9 IU/dL for FVIII activity. Increase was higher in blood group non-O.	Biguzzi et al., 2021 [69]

## 5. Impact of Aging on von Willebrand Disease

The normal reference range of plasma VWF levels is between 50 and 200 IU/dL in a healthy population [87]. However, as described in the previous section, quantitative (Types 1 and 3) and qualitative (Type 2) deficiencies of VWF occurs that result in von Willebrand disease (VWD), characterized by excessive mucocutaneous bleeding. Although VWF levels in the range of 30–50 IU/dL have been generally considered as an indication of VWD, currently this range is considered as a low normal VWF level, which is nevertheless a risk factor for developing bleeding disorders [87]. The observation of an age-associated increase in plasma VWF levels of a healthy population suggests that aging may also impact elderly VWD patients. While Type 2 and Type 3 VWD are diagnosed due to qualitative deficiencies or severe reductions in VWF levels, diagnosing and differentiating mild forms of VWD—Type 1 from low levels of VWF is more challenging. Regardless of classification into normal but low levels, or Type I mild VWD, this population might be a target group that could benefit from the age-related rise in VWF levels. Consistent with this hypothesis, VWF levels were shown to rise with aging in milder forms of VWD, but not in severe forms, such as Type 2 and 3 VWD [22,88,89]. Furthermore, normalization of VWF levels with aging may affect symptoms, severity, and diagnosis, in mild forms of VWD. A retrospective study of 126 patients with Type 1 VWD or ‘low VWF’ indicated a significant increase in VWF levels and activity in elderly VWD Type 1 patients, leading to complete VWF levels normalization in 27.8% of participants, although amelioration of bleeding symptoms was not shown [90]. Another retrospective study of Type 1 VWD patients aged 16–60 for eleven years found normalization of VWF antigen levels, activity, and FVIII levels in 18 patients out of 31 [91]. Additionally, Willebrand in the Netherlands (WiN-) study of 664 VWD patients aged 16–85 years found that with aging VWF levels fell within the normal reference range in mild Type 1 VWD, while remained unchanged in Type 2 VWD [92]. Increased VWF levels were also followed by elevated VWF activity and FVIII levels in Type 1 VWD, but not in Type 2 VWD [92]. Very few studies have been performed involving Type 2 VWD patients, whereas none of the studies have included Type 3 VWD patients. This may reflect the expectation that qualitative deficiencies associated with Type 2 VWD will not be ameliorated with increased levels; and that the nature of extensive deficiency associated with Type III VWD (i.e., gene deletion), is incompatible with potential increases in VWF levels.

To evaluate the effect of comorbidities in VWD patients, a cross-sectional study was performed that included 536 VWD Type 1 and Type 2 patients aged 16–83 year [22]. It was found that hypertension, diabetes mellitus, thyroid dysfunction and cancer were associated with increased VWF levels in Type 1 VWD compared to VWD patients without these comorbidities, whereas no association was found in Type 2 VWD patients [22]. Lowered VWFpp:VWF-antigen ratio indicated that reduced clearance of VWF in circulation was a potential cause of elevated VWF levels in these comorbidities [22]. It is vital to investigate and establish the effect of aging on VWD patients since it has serious consequences on the treatment of VWD generally, as well as in conjunction with other age-related comorbidities.

A recent study has described in details the impact of aging on VWF levels and presented an in-depth discussion of its association with bleeding symptoms in VWD patients [89]. However, despite accumulating evidence in support of age-associated increase in VWF levels and function, whether this results in amelioration of bleeding symptoms in VWD patients remains unclear. The uncertainty and conflicting reports are proposed to be due to differences in the study designs and procedures, patients’ characteristics, or presence of comorbidities. It is also hypothesized that with aging there may be a requirement for higher levels of VWF for optimal hemostasis; thus, amelioration of bleeding symptoms may not be observed in aging VWD patients [89].

## 6. Potential Mechanisms of Age-Associated Increase in VWF

An increase in plasma levels of VWF may result from increased regulated and/or constitutive secretion. Numerous factors including thrombin, histamine, nitric oxide (NO), calcium, vasopressin, epinephrine, leukotrienes, superoxide anions, sphingosine-1-phosphate, endothelin-1 (ET-1), prostacyclin and shear stress were shown to affect VWF plasma levels [93,94,95,96,97,98]. While some factors may alter the regulation of VWF secretions, others may affect VWF levels by altering VWF gene transcription/post-transcriptional activity [95,96,97]. A combination of alterations in secretion and production levels may contribute to increase circulating VWF levels.

Although the underlying mechanism of age-associated increase in VWF remains unclear, age-related alterations in the circulatory and vascular physiology/pathophysiology could provide some insights. Elevated VWF levels with aging may be an intrinsic property of aged endothelial cells, may occur in response to various age-related vascular dysfunctions, and/or increased inflammation. Aging generally leads to the development of a pro-inflammatory phenotype in the elderly population [99,100], and inflammation is a major contributing factor to increased thrombogenicity [101]. Pathogenicity of inflammation-induced thrombosis is complex, and interactions between inflammation and thrombosis is a bidirectional process: inflammation promotes platelets aggregation and coagulation, while activated platelets and coagulation factors promote inflammation [99,100]. Vascular injury is the key factor that associates inflammation with thrombosis; however, inflammation-induced thrombosis may also develop without endothelial injury [100]. Elevated levels of pro-inflammatory markers have been reported in many older individuals, even in the absence of pre-existing health conditions or related risk factors [102,103]. Observation of elevated inflammatory mediators, leading to chronic low-grade inflammation in the aged population, is known as “inflammaging”; and it promotes endothelial dysfunction [102]. VWF plasma level is known as a useful marker for endothelial dysfunction and may be relevant to the disease process. An age-related alteration in endothelium and surrounding microenvironment may also contribute to age-associated increase in VWF levels and thrombogenicity. Aging alters the levels of vasodilator NO and vasoconstrictor endothelin1 (ET-1), which are both regulators of VWF secretion [104]. Nitric oxide (NO) inhibits exocytosis of WPB and platelets’ alpha granules; hence, it inhibits VWF release [93,105]. Aging is shown to result in decreased NO bioavailability and sensitivity to NO, thus ameliorating/reducing its’ inhibitory effect on exocytosis and consequently promoting increased VWF secretion from storage organelles [106,107]. In contrast, ET-1 that increases VWF expression in endothelial cells, and shown to cause elevated circulating VWF levels when infused in men [108], exhibits an elevated levels with aging.

Aging also affects circadian rhythm and is associated with fragmentation or loss of rhythm [106]. VWF promoter activity is under direct regulation of major circadian transcriptional regulators CLOCK/Bmal1 [109]. Bmal1 deficiency not only directly increases VWF mRNA and protein levels, but also downregulates nitric oxide synthase (eNOS) activity and consequently decreases NO production; thus, indirectly contributing to VWF release as well [109,110]. However, it should be noted that the reported age-associated decrease in Bmal1 expression was based on analyses of the master circadian pacemaker in the suprachiasmatic nucleus (SCN), the hippocampus, and cingulate cortex [111]. Further studies are required to determine whether aging similarly downregulates Bmal1 in other tissues/cell types, including endothelial cells to directly establish that decreased Bmal1 with aging increases VWF levels.

Another potential contributor to the age-related increase in VWF could be related to alteration in osmolarity. In-vivo studies in mice demonstrated that water restriction and elevated plasma sodium concentration, above normal physiological range, significantly increased VWF mRNA and protein levels [112]. This process was shown to be mediated through binding of the tonicity-regulated transcription factor NFAT5 to the VWF promoter [112]. With aging water intake decreases and the risk of dehydration increases. Numerous studies on the elderly have reported that aging decreases perceived thirst and water intake in response to hypovolemia, hypertonicity, and hormonal stimulus [113,114,115]. Thus, dehydration in older people increases sodium concentration and this could lead to an increase in VWF production as well as secretion.

In addition to increased VWF levels, aging is also reported to be associated with a decrease in the level and activity of VWF cleaving enzyme ADAMTS13 [67,72]. Thus, a combination of these two events cumulatively contributes to the generation and persistence of ultra-large VWF multimers in circulation [67,72]. Under the condition of high shear stress, ADAMTS13 mediates increased proteolysis of high molecular weight (HMW) [116], but aging is reported to be also associated with reduced shear stress, thus further contributing to a condition that promotes persistence of high molecular weight VWF in the circulation and elevated platelets aggregation [117]. Together these age-associated alterations directly and/or indirectly increase VWF production, secretion, and stability, potentially leading to the generation of platelets aggregates and consequently increased thrombotic events (Figure 2).

## 7. VWF and COVID-19

Various infectious diseases including HIV [118], malaria [119], scrub typhus [120], and Dengue virus [121] infection have been associated with thrombosis and increased circulating VWF levels. The COVID-19 disease caused by the SARS-CoV-2 (CoV-2) virus starts with an early infectious stage, but it may be followed by viral pneumonia and systemic inflammation that could potentially lead to respiratory failure and multiple organ dysfunction [122,123,124,125]. It is now well established that the COVID-19 is also associated with hypercoagulation and severe thrombotic complications—‘Corona virus-associated coagulopathy’ (CAC) [126,127,128]. Several studies have reported incidence of ischemic stroke, deep vein thrombosis, pulmonary thromboembolism, and thrombosis in small brain vessels leading to cerebral microbleeds in 20–30% of COVID-19 patients admitted to the intensive care unit (ICU). The incidences of these coagulopathies are significantly higher among aging population of COVID-19 patients and encompass more than 50% of elderly non-survivors [129,130,131,132]. CoV-2 infection is reported to facilitate vascular inflammation (vasculitis) and enhance the prothrombotic state, potentially leading to organ dysfunction due to the presence of microangiopathy and microthrombi in vascular beds of multiple organs [133,134,135].

Multiple hematological abnormalities are observed in COVID-19 patients including elevated levels of fibrinogen and D-Dimer in critically ill patients [127,136,137]. However, hematological analyses of these patients have also found that VWF levels and activities were elevated five to six folds beyond the upper normal limit [128,138]. In a cohort of 150 elderly COVID-19 patients with a medical history of cardiovascular, respiratory, and multiple organ abnormalities, median VWF levels were reported at 455% of normal [128]. In some cases, VWF elevation is augmented by a deficiency of the VWF cleavage enzyme ADAMTS-13 or reduced ADAMTS13 activity in the blood [126,139,140]. Considering the age-associated increase in VWF, it is plausible to hypothesize that elevated basal levels of VWF in elderly may increase their susceptibility towards development of CAC complications as a result of CoV-2 infection. Consistent with this hypothesis are the observations that microangiopathy in the lung is highly prevalent among COVID-19 patients with CAC [125,141,142,143], and that age-associated increase, specifically in cellular expression of VWF, is observed in lung microvasculature [82].

CoV-2 infection is not only associated with increased plasma levels of VWF but also was reported to have a qualitative impact on the VWF activity, specifically enhancing its adhesive properties, such as collagen binding [140]. While more investigation is required, emerging data suggest that COVID-19 is associated with increased consumption of HMWM VWF [140,144,145], which could be indicative of increased platelets aggregte formation. Consistent with this, lower platelet counts were observed in some COVID-19 patients, which could be due to hyperactivation of platelets and aggregate formation following platelet interaction with VWF [128,146].

Regarding ADAMTS13 levels in COVID-19 patients, the reports are less consistent. Some studies reported normal levels of ADAMTS13 while few reported decreased levels of ADAMTS13[126,147,148]. There have also been reports of a significantly reduced ratio of ADAMTS13/VWF, which suggest that while ADAMTS13 may or may not be reduced, a very high concentration of VWF could overwhelm the capacity of ADAMTS13 [145]. It is also noteworthy that neutrophil extracellular traps (NETs) are found in the microvascular thrombi present in the heart, lung, or kidney of COVID-19 patients [149]. It is reported that contents released from NETs might reduce enzymatic activity of ADAMTS13, consequently increasing VWF levels [150]. Summary of the literature reports of investigation into the levels and activities of VWF and ADAMTS13 in critically ill COVID-19 patients with and without coexisting comorbidities is presented in Table 2. While the mechanism for increased VWF levels in COVID-19 patients with the hypercoagulable state remains unclear, it is noteworthy that elevated VWF levels are predominantly reported in critically ill patients with health conditions/comorbidities that are also associated with increased VWF levels, including elderly and/or those with hypertension and diabetes mellitus. Elevated VWF levels associated with various chronic diseases and natural aging may generate a heightened prothrombotic state. This, combined with CoV-2 induced increase in VWF levels, could significantly tip the balance of pro- and anti- thrombotic state in favor of thrombosis.

CoV-2 may directly infect endothelial cells, and/or indirectly stimulate endothelial cells to release VWF as a result of enhanced inflammatory environment. The lungs were reported as the main target of CoV-2, which can bind to Angiotensin-converting enzyme 2 (ACE2), on the alveolocytes [151]. ACE2 receptor, which belongs to the renin-angiotensin system and has a major role in regulating blood pressure, is also expressed on the endothelial surfaces of the heart, lung, kidney, and systemic vessels [152]. ACE2 is proposed to mediate CoV-2 entry into endothelial cells [153,154], independently, or potentially in combination with other proteins, which have been also identified as receptors for CoV-2 on endothelial cell surface [155]. However, interaction of virus particles with ACE2 may also lead to a loss of ACE2 enzymatic function and consequently decrease plasma level of vaso-protective angiotensin [156,157,158]. This in turn could result in acute-phase inflammatory responses leading to endothelialitis (vascular endothelial inflammation) and hypersecretion of VWF [159,160,161]. Proinflammatory conditions arising as a result of cytokine storm generation further contribute to prolonged endothelial stimulation and activation of complement cascade; both of which stimulate VWF release [159,160,161]. Inflammation, endothelial injury, and dysregulated immune response instigate assembly of membrane attack complex (MAC) on endothelial cells, which increases endothelial cytosolic Ca^2+^ concentration that stimulates VWF release from Weible-Palade bodies [162]. Inflammatory cytokine TNF-alpha can regulate VWF expression indirectly by decreasing NO synthesis through inhibition of eNOS expression in endothelial cells [163,164]. Consistent with this, significantly decreased NO levels has been reported as a result of CoV-2 infection [165,166]. Additionally, significantly increased IL-8 and IL-6 inflammatory cytokines also promote VWF secretion in a concentration-dependent manner. Furthermore, IL-6 has been reported to also decrease ADAMTS13 activity, which further increases accumulation of ultra-large VWF levels with higher activity [167].

**Table 2 jcm-10-04190-t002:** VWF and ATDAMTS13 levels ad activity reported in critically ill COVID-19 patients with and without preexisting comorbidities. Normal reference VWF antigen range (%) 42–136 ^a^. Normal reference VWF activity range (%) 42–168 ^b^. Normal ADAMTS13 antigen reference range (%) 40–130 ^c^. Normal ADAMTS13 activity reference range (%) 50–150 ^d^. The value is reported as median (interquartile range) ^e^.

Location	Patient Characteristics (n)	Mean Age (years)	VWF Antigen (%) ^a^ Mean (Min–Max)	VWF Activity (%) ^b^ Mean (Min–Max)	ADAMTS13 (%) ^c^ Mean (Min–Max)	ADAMTS13 Activity (%) ^d^ Mean (Min–Max)	Author
Italy	Intubated COVID-19 patients (11)	–	529 (210–863)	387 (195–550)	–	–	Panigada et al., 2020 [146]
The Netherlands	COVID-19 patients admitted to ICU (12)	61.8	408.0	374.0	48.0	–	Huisman et al., 2020 [168]
France	Patients with radiological signs of interstitial pneumonia (212)	63.9	361.0	–	–	–	Rauch et al., 2020 [169]
Germany ^e^	COVID-19 patients with high severity (150)	63.0	455 (350–521)	328 (212–342)	–	–	Helms et al., 2020 [128]
Italy ^e^	COVID-19 patients with high severity (19)	59.0	476 (381–537)	388 (328–438)	–	55 (42–68)	Mancini et al., 2021 [144]
United Kingdom ^e^	Intubated ICU patients (24)	65.0	350 (302–433)	–	–	–	Ladikou et al., 2020 [170]
Ireland	COVID-19 patients admitted to ICU (28)	55.0	690.2 (467–848.4)	–	–	–	Ward et al., 2021 [171]
France ^e^	Patients with critical form of COVID-19 in ICU (89)	62.0	507 (428–596)	399 (333–537)	–	–	Philippe et al., 2021 [172]
France ^e^	COVID-19 patients with Venous thromboembolic events (38)	63.0	522 (411–672)	–	–	59 (38.8–70.5)	Delrue et al., 2021 [173]
Italy	Intubated ICU patients (6)	62.0	634 (455–772)	450 (339–496)	37.3 (24–56)	–	Morici et al., 2020 [148]
United States	Intubated ICU patients (48)	64.0	565.0	390.0	–	–	Goshua et al., 2020 [138]
Italy	Patients with COVID-19 pneumonia (37)	61.8	280.8	265.1	–	–	Taus et al., 2020 [174]
Spain	ICU patients with the history of hypertension and diabetes mellitus (22)	68.0	368.6	–	–	38.9	Marco et al., 2021 [175]
Italy ^e^	Patients with novel coronavirus pneumonia (10)	61.0	324.1	341.5	69.0	–	De Cristofaro 2021 [176]
Spain ^e^	COVID-19 patients with cardiovascular disease and diabetes (23)	64.0	306.0	–	47.3	–	Blasi et al., 2020 [177]
Germany	COVID-19 patients with mild to moderate severity (75)	66.0	403.0	–	67.8	–	Doevelaar et al., 2021 [145]
Italy	Patients with the history of hypertension and diabetes (19)	69.0	331.4	321.7	–	–	Ruberto et al., 2020 [178]
United States ^e^	COVID-19 non-survivors (90)	72.5	441.0	321.0	–	48.8	Sweeney et al., 2020 [179]
Spain ^e^	Severe COVID-19 patients (50)	–	355 (267–400)	–	53.2 (38.8–65.3)	–	Rodríguez et al., 2021 [180]
Italy	COVID-19 non-survivors (9)	72.0	395.5	–	32.2	–	Bazzan et al., 2020 [126]
Belgium	COVID-19 patients admitted to ICU (9)	57–64	475.0	429.0	45.0	–	Hardy et al., 2020 [181]
United States ^e^	17	42–58	448 (362–529)	313 (190–347)	–	–	Masi et al., 2020 [182]
Sweden ^e^	Patients with COVID-19 high care (12)	53–69	425 (321–465)	–	57 (42–62)	–	Meijenfeldt et al., 2021 [183]
Germany	COVID-19 non-survivors (5)	78.0	260.4	217.6	–	43.3	De Jongh et al., 2021 [184]
France ^e^	COVID-19 patients admitted to ICU (22)	–	456 (402–493)	355 (297–416)	458 (364–615)	–	Pascreau et al., 2021 [185]

Another factor that may also contribute to elevated levels of VWF in COVID-19 patients is development of hypoxic conditions. COVID-19 patients may suffer from acute lung injury that causes acute respiratory distress syndrome (ARDS) and decreases tissue oxygenation, leading to hypoxic condition. Hypoxia was shown to be a mediator of VWF release from WPB [186]. Furthermore, we have previously demonstrated that hypoxia also induces VWF transcriptional upregulation, and consequently contributes to increased platelet aggregates formation [187,188]. Specifically, in the lungs of mice, hypoxia altered the VWF expression pattern, leading to expression of VWF in a significant proportion of lung microvascular endothelial cells that generally do not exhibit VWF expression under normoxic conditions [187]. Thus, COVID-19 induced hypoxia may upregulate VWF levels, and this has been supported by the clinical observation of rising VWF levels with increased oxygen requirements in a few COVID-19 patients [169]. Additionally, it is noteworthy that elevated VWF levels has been reported only in COVID-19 positive pneumonia patients [176]. This suggests that synergistic effects of hypoxia with other COVID-19 related conditions might be responsible for increase in VWF levels during the disease progression. Thus, pre-existing factors, such as aging and comorbidities, in combination with hypoxic, immunologic, and proinflammatory alterations that ensue in COVID-19 infection, may collectively contribute to a significant increase in VWF levels and consequently increased thrombogenicity.

COVID-19 does not affect all individuals in a similar manner, and this is also reflected in development of hypercoagulability and thrombosis. This, however, is consistent with highly variable basal levels of VWF in population [35], which may play a role in predisposing certain individuals to increased risk of CAC development. For instance, individuals with high basal levels of VWF, when aged and/or have comorbidities, if infected with CoV-2 may reach a VWF threshold that leads to the development of thrombosis, while people with low basal levels of VWF may not. Consistent with this hypothesis, some studies have indicated that COVID-19 patients with Blood Group O (25% lower VWF levels than other blood groups) are significantly underrepresented in thrombogenic complications [189,190]. Similarly, Type I VWD patients with significantly lower VWF levels are also underrepresented in groups of COVID-19 patients with thrombogenic consequences. These observations suggest that determining the level and activity of VWF in COVID-19 patients may provide a useful prognostic marker towards COVID-19 morbidity and mortality as well as identifying patient populations that are at the risk of developing thrombotic complications.

## 8. Conclusions

Aging is associated with increased risk of arterial, venous, and microvascular thrombosis in the elderly population. Elevated levels of VWF in circulation, as well as cellular levels, are also reported with aging. Considering the central role of VWF in thrombus formation, a likely association between this rise in VWF and thrombogenicity is conceivable. Various external factors, specifically alterations in inflammatory and immunological landscape may significantly contribute to the increased production of VWF with aging. However, age-associated increase in VWF occurs in healthy population and exhibits a gradual constant increase. Thus, it is also conceivable that inherent alterations in endothelial cell phenotype may occur with aging that culminates in increased production of VWF. Such possibility may be specifically important in relation to microvascular endothelial cells, which do not normally exhibit VWF expression. Testing of this hypothesis will provide significant insights towards determining the underlying molecular basis of age-associated increase in VWF. It will also contribute to potential development of targeted therapies to combat various age-associated macro- and microvascular diseases in elderly population with and without comorbidities.

Therapeutic approaches that are developed or under consideration include strategies to interfere with binding of platelets to VWF, or reduction in the levels of ULVWF multimers. For instance, some clinical treatment options include blocking of VWF and platelet interaction using anti VWF antibody such as AJW200 [191,192,193] or a humanized anti-VWF antibody fragment (nanobody) caplacizumab, which bind to VWF A1 domain [194,195]. In addition, anfibatide, a snake-venom derived compound that is an antagonist of GPIb receptor on platelets, can inhibit platelet-VWF interaction [196,197]. Reduction in the levels of ULVWF may be achieved through restoration/increase of ADAMTS13 function by introduction of recombinant ADAMTS13 (rADAMTS13) [198] or through the action of N-acetylcysteine (NAC) that disrupts disulfide bonds in the VWF A1 domain [199,200]. In addition, aptamers that are DNA or RNA single-stranded nucleic acids, which bind to their target protein with high affinity and specificity, could present as novel potential therapeutic agents. DNA aptamer TAGX-0004 [201] and RNA aptamer BT200 [202] are inhibitors of the VWF A1 domain that may be considered as potential inhibitors of VWF function.

## Figures and Tables

**Figure 1 jcm-10-04190-f001:**
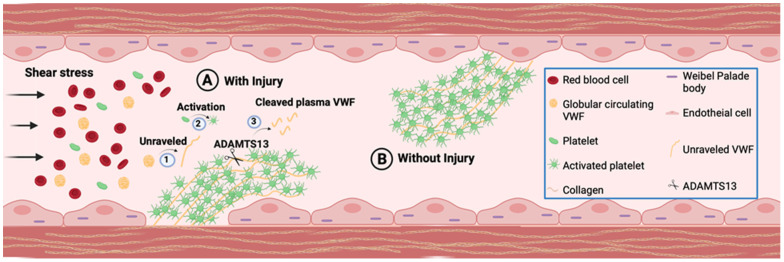
**VWF Function.** VWF is stored in Weibel-Palade bodies (WPB) and deposited in extracellular matrix of vascular endothelial cells, as well as circulating in plasma as globular form. (**A**) At the site of injury in presence of shear stress globular VWF attach to the exposed subendothelial collagen layer and (1) unravel to form the most adhesive form of VWF. The extracellular matrix deposited VWF is similarly exposed to bind platelets. Platelets captured by unraveled VWF are (2) activated and form VWF-platelet aggregates to seal the damaged vessel wall and prevent blood leakage while repair. Following the completion of repair process ADAMTS13 (Scissors) will (3) cleave UL VWF to smaller VWF multimers to resolve platelet aggregates. (**B**) External stimuli induce secretion of VWF, which may bind to endothelial luminal surface membrane and mediate platelet aggregate formation through a similar process as that observed at the site of vascular injury.

**Figure 2 jcm-10-04190-f002:**
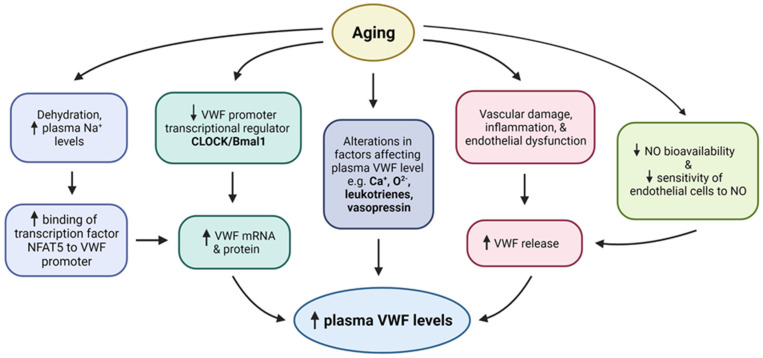
**Potential mechanisms of age associated increase in VWF levels.** Upward and downward arrows depict increase and decrease respectively.

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
