# Peer review of "Age-Associated Increase in Thrombogenicity and Its Correlation with von Willebrand Factor"

_jcm, 2021, doi:10.3390/jcm10184190_

Round 1
Reviewer 1 Report
In this review, the authors consider the relationship between aging and von Willebrand Factor (vWF) levels, highlighting the putative molecular mechanisms undergo this correlation. This article gives an interesting scientific summary on a field that has been studied for decades. Lately, the interesting association between vWF and Covid-19 was considered.
The authors comprehensively analyzed and discussed the data considered, and the considerations reported are generally well written and easily understood.
Author Response
We are very grateful for the positive and encouraging comments of the reviewer and thank the reviewer for the time and effort to review our manuscript.
Reviewer 2 Report
Review of “Age-associated increase in thrombogenicity and its correlation with von Willebrand factor”
This review article summarizes the literature with respect to changes in vWF physiology associated with aging and hypothesizes about its role in hypercoagulability in the aged population. The review is comprehensive and well written.
Major Comments:
- None.
Minor Comments:
- Page 6, line 268 – “the nature of extensive deficiency associated with type III VWD (i.e gene deletion), is not likely to be permissible to alterations in VWF levels.” Does not make sense, wrong verb?
- Page 2, line 86 – should “consecutive” be “constitutive?”
Author Response
We thank the reviewer for the positive and encouraging review and are grateful for constructive suggestions, which we have addressed as follows:
The sentence to which the reviewer has referred as being grammatically incorrect is now modified in the revised manuscript to “the nature of extensive deficiency associated with type III VWD (i.e gene deletion), is incompatible with potential increases in VWF levels.” Also the word consecutive has been corrected to constitutive.
Reviewer 3 Report
This is a nicely written review about age-associated increase in von Willebrand factor and the various pathomechanisms behind it.
I just have a couple of minor comments:
1) Just out of curiosity: With age, vWF levels increase which is accompanied by a simultaneous increase in FVIII. Does this also translate into less bleeding in mild hemophilia A?
2) The section about vWF in Covid-19, although it ties in with recent events, is interesting but does not fit the overall topic of the review.
3) Optional: an additional table summarizing the mentioned trials of vWF levels (non-Covid) could be helpful.
4) Optional: a short outlook on future treatment possibilities would be interesting. E.g. recently, an aptamer targeting vWF (BT200), has been developed. (Kovacevic et al) Further investigational treatment options include AJW200, rADAMTS13, Anfibatide and caplacizumab.
5) spell check required
Author Response
We thank the reviewer for the positive and encouraging review and are grateful for constructive suggestions, which we have addressed as follows:
- Regarding whether age-associated increased VWF translates into less bleeding in mild hemophilia A, we have now added (page 5 revised manuscript) the following paragraph and references in this regard.
“This increase in VWF levels was followed by simultaneous elevation in FVIII coagulant activity and antigen levels (reaching up to 1.5-fold by later life in healthy individuals) [68,69]. A recent cross-sectional study of 2923 individuals also reported elevation of VWF:Ag as well as FVIII coagulation activity per decade of age after adjustment for comorbidities including body mass index, inflammation and hormone use; and the increase was higher for non-O blood group individuals [69]. However, the influence of aging and plasma VWF levels on FVIII level and coagulant activity in mild Hemophilia A patients is scarcely studied. One study reported a significant positive correlation between FVIII coagulant activity and aging without amelioration of bleeding [70]; whereas another study identified only minor influence of aging on FVIII levels/activity [71].”
- We appreciate the reviewer’s comment that the COVID-19 may not completely fit the overall topic of the review. However, we were considering the reports of increased VWF in COVID19 patients and that specifically elderly covid19 patients are presenting with thrombotic complications. Thus we believed that a discussion of this topic could be potentially relevant to age-related increased VWF levels that could exacerbate the increase in thrombogenicity in COVID19 elderly patients.
- An additional table summarizing the mentioned trials of VWF levels (non-Covid) has now been included as table 1. We thank the reviewer for this suggestion.
- A short outlook on future treatment possibilities are now included at the end of conclusion section. We thank the reviewer for the excellent suggestion and are grateful for the information and references that the reviewer had kindly provided to guide us.
- We have tried our best to review the manuscript for spelling and editorial corrections which are tracked in the revised manuscript.